# SubMix: Practical Private Prediction for Large-scale Language Models

## Abstract

Recent data-extraction attacks have exposed that language models can memorize some training samples verbatim. This is a vulnerability that can compromise the privacy of the model's training data. In this work, we introduce SubMix: a practical protocol for private next-token prediction designed to prevent privacy violations by language models that were fine-tuned on a private corpus after pre-training on a public corpus. We show that SubMix limits the leakage of information that is unique to any individual user in the private corpus via a relaxation of group differentially private prediction. Importantly, SubMix admits a tight, data-dependent privacy accounting mechanism, which allows it to thwart existing data-extraction attacks while maintaining the utility of the language model. SubMix is the first protocol that maintains privacy even when publicly releasing tens of thousands of next-token predictions made by large transformer-based models such as GPT-2.

## 1 Introduction

The advent of transformers (Vaswani et al., 2017) has fostered a dramatic advancement in the capabilities of generative neural language models (LMs), enabling large-scale models such as GPT (Radford et al., 2019; Brown et al., 2020) to generate realistic, human-like text. Unfortunately, these impressive capabilities also come at a cost to privacy, as the amount of excess parameters in the LM enables it to memorize certain training samples. Consequently, several recent works have demonstrated practical training-data extraction attacks that reproduce entire sentences from the training dataset verbatim by querying the LM as an API (Carlini et al., 2019; 2020). These attacks expose the privacy risks of large-scale LMs, especially when their training data contains sensitive information such as addresses and personal ID numbers.

Existing solutions to data extraction attacks focus on using differential privacy (DP; Dwork et al. (2014)), which provably protects against privacy attacks (Yeom et al., 2018). Techniques such as DP-SGD (Abadi et al., 2016) have been applied to train differentially private neural networks on both vision and language tasks (McMahan et al., 2017; Papernot et al., 2018). However, the threat model in DP-SGD implicitly assumes that the adversary has full access to the private model's parameters and gradients during training, which results in pessimistic information leakage bounds that are unreasonable for most models. Indeed, existing work only performs DP-SGD training of small feedforward networks (Kerrigan et al., 2020) and RNNs (McMahan et al., 2017; Ramaswamy et al., 2020), often with an unsatisfactory privacy-utility trade-off. Training large machine-learning models with DP-SGD remains an open challenge (Jayaraman & Evans, 2019; Tramèr & Boneh, 2020).

Our study deviates from prior work by, instead, considering the problem of *private prediction* (Dwork & Feldman, 2018) using non-private language models fine-tuned on a private corpus. We propose SubMix, a novel private prediction mechanism for answering next-token queries. Focusing on private prediction affords SubMix three notable advantages: (1) Private prediction does not require modification of the training algorithm, which makes use of large-scale LMs feasible. (2) Private prediction allows us to leverage the probabilistic nature of next-token sampling for highly efficient privacy accounting. (3) Private prediction allows us to leverage public pre-trained LMs[1] to obtain private predictive distributions that do not require noise addition to privatize the model's predictions.

SubMix utilizes an ensemble of LMs fine-tuned on disjoint parts of the private corpus and privatizes predictions by mixing the next-token distribution with that of a public pre-trained LM. The mixing weight is adaptively tuned based on the degree of consensus among models in the ensemble. If

---

[1]We do not provide privacy guarantees or text extraction protection for the public corpus on which the LMs are pre-trained; our privacy guarantees only on apply to the private corpus on which the LM is fine-tuned.

all models predict the same next-token distribution, then it is impossible for the next token to leak sensitive information about any unique individual so no mixing is required. By contrast, if models in the ensemble have high disagreement, SUBMIX will mix predictions with those of the public pre-trained model to minimize privacy leakage. This allows SUBMIX to perform accurate next-token prediction for most queries while preserving the privacy of the private corpus.

For any sequence of next-token queries issued to SUBMIX, we measure the amount of privacy leakage in the response using Rényi divergence (Rényi, 1961). Our privacy notion, which we refer to as *operational privacy*, is a sufficient condition for preventing samples that are unique to any user from being generated by the SUBMIX mechanism. Importantly, operational privacy allows us to perform tight data-dependent privacy accounting to upper bound the privacy loss of SUBMIX when answering a variable-length query sequence. Concretely, when answering up to $1,024$ next-token queries, SUBMIX realizes nearly $75\%$ of the perplexity improvement that non-private fine-tuning would have achieved on GPT-2 models (Radford et al., 2019), with privacy leakage as small as $\epsilon = 2$. We also show that SUBMIX can effectively prevent existing data extraction attacks against GPT-2.

## 2 PROBLEM FORMULATION

We begin by setting up the problem of private next-token prediction and reviewing existing literature on differential privacy. We then define and discuss the notion of operational privacy for SUBMIX.

### 2.1 PRELIMINARIES

Let $\Sigma$ denote a fixed finite vocabulary set. We use lower-case letters to denote single tokens (such as $x \in \Sigma$) and use bold font to denote contexts or strings of tokens (such as $\mathbf{x} \in \Sigma^*$). A (causal) language model $h$ is a mapping from *context strings* to a distribution over next tokens: $h : \Sigma^* \to \boldsymbol{\Delta}^{|\Sigma|}$, where $\boldsymbol{\Delta}^{|\Sigma|}$ is the $|\Sigma|$-dimensional probability simplex. For a particular context $\mathbf{x} \in \Sigma^*$, let $h(\mathbf{x})$ denote the next-token distribution vector obtained from evaluating $h$ on context $\mathbf{x}$, and let $h(z|\mathbf{x}) \in [0, 1]$ denote the probability mass on a token $z \in \Sigma$.

**User-level Corpus**  Let $\mathcal{D}$ denote a dataset of unstructured text, which is a set of token sequences $\mathbf{x} \in \Sigma^*$. We assume that $\mathcal{D}$ is generated by a set of $n$ distinct users, each holding a subset $\mathcal{D}_i$ of the full dataset $\mathcal{D}$, *i.e.*, $\mathcal{D} = \bigcup_{i=1}^{n} \mathcal{D}_i$ with $\mathcal{D}_i \cap \mathcal{D}_j = \emptyset$ for $i \neq j$. We refer to each $\mathcal{D}_i$ as a *user-level corpus* for user $i$. As a concrete example, in the context of social media posts, $\mathcal{D}_i$ would contain *all* of the non-public posts made by user $i$. We aim to provide privacy guarantees for a model that is non-privately fine-tuned on the dataset $\mathcal{D}$.

**Next-token Prediction**  One popular use case for language models is to perform *next-token prediction*, that is, return a token $z$ when queried with a context $\mathbf{x}$. Such a query-answering API is useful for applications such as smart keyboard for auto-correction and text completion (Mirowski & Vlachos, 2015; Hertel, 2019). Typical approaches for next-token prediction involve sampling $z$ from the next-token distribution vector $h(\mathbf{x})$; see Holtzman et al. (2019). Large transformers trained on unstructured internet text have achieved remarkable success for this task, producing natural-looking sentences via sequentially generating next tokens from a given prompt (Brown et al., 2020).

**Text Extraction Attacks**  Recent studies have shown that it is possible for next-token prediction APIs to reveal sensitive private information contained in the training dataset. Carlini et al. (2019) defined $\kappa$-*eidetic memorization* to formalize the notion that extraction of strings that are uncommon in the corpus can lead to violations of user privacy.

**Definition 2.1** ($\kappa$-eidetic memorization (Carlini et al., 2019))**.**  A string $s$ is $\kappa$-eidetic memorized by an LM $h$ if $s$ is *extractable*[2] from $h$ and $s$ appears in at most $\kappa$ examples in the training data $\mathcal{D}$.

Carlini et al. (2019) showed that if the training dataset contains token sequences of the form: "My social security number is □□□–□□–□□□□" where □ represents a digit of a user's

---

[2]Carlini et al. (2019) define *text extraction* informally. In the supplement, we formalize it within the framework of statistical hypothesis testing and show that differential privacy is sufficient to prevent eidetic memorization.

social security number (SSN), then it is subtantially more likely for the LM trained on $\mathcal{D}$ to generate the exact SSN appearing in $\mathcal{D}$ compared to a random SSN. As a result, it is possible to design an efficient *extraction attack* that reproduces such unique sequences in the training dataset. Carlini et al. (2020) further extended this attack to large transformer-based LMs such as GPT2 (Radford et al., 2019), extracting memorized personal information such as name and address contained in the model's training dataset. Motivated by these shortcomings, this paper studies notions of privacy that can prevent such text-extraction attacks while preserving the model's utility.

## 2.2 DIFFERENTIAL PRIVACY

Differential privacy (Dwork et al., 2014) is a powerful mathematical framework for privacy-preserving data analysis. The underlying principle in differential privacy and all its variants is the notion of *indistinguishability*. Informally, a mechanism $\mathcal{M}$ is private if, given two adjacent datasets $\mathcal{D}$ and $\mathcal{D}'$, the mechanism's outputs $\mathcal{M}(\mathcal{D})$ and $\mathcal{M}(\mathcal{D}')$ are approximately indistinguishable. Hence by observing the output of $\mathcal{M}$, it is difficult for an adversary to discern the difference between $\mathcal{D}$ and $\mathcal{D}'$. The above informal definition of privacy can be made mathematically precise by specifying: (1) the notion of adjacency between datasets $\mathcal{D}$ and $\mathcal{D}'$, and (2) the notion of approximate indistinguishability.

**Differentially Private Training** Prior work on private LM training (McMahan et al., 2017; Ramaswamy et al., 2020) adopted the definition of *user-level adjacency*: $\mathcal{D}$ and $\mathcal{D}'$ are adjacent if they differ in a single user's data. Approximate indistinguishability is defined in terms of divergences and is applied to the trained model: The private training algorithm $\mathcal{M}(\mathcal{D})$ induces a distribution over models, and indistinguishability requires that $D(\mathcal{M}(\mathcal{D})||\mathcal{M}(\mathcal{D}')) < \epsilon$ for some divergence $D$ and small constant $\epsilon > 0$. Popular choices include the *max divergence* (Dwork et al., 2014) and the *Rényi divergence of order* $\alpha$ (Rényi, 1961):

$$D_\infty(P||Q) = \sup_{x \in \text{supp}(Q)} \log P(x) - \log Q(x), \qquad D_\alpha(P||Q) = \frac{1}{\alpha - 1} \log \mathbb{E}_{x \sim Q} \left[ P(x)/Q(x) \right]^\alpha.$$

Specializing to the choice of Rényi divergence, we define *user-level Rényi differential privacy* (RDP; Mironov (2017)) for private training as follows.

**Definition 2.2** (User-level RDP for private training). For $\alpha > 1$, let $D_\alpha$ denote the order-$\alpha$ Rényi divergence. A private training algorithm $\mathcal{M}$ is an $(\alpha, \epsilon)$-RDP mechanism if for any $\mathcal{D}$ and $\mathcal{D}'$ that differ in only one user's data $\mathcal{D}_i$, we have $D_\alpha(\mathcal{M}(\mathcal{D})||\mathcal{M}(\mathcal{D}')) \leq \epsilon$.

In order to satisfy the criteria in Definition 2.2 for neural language models, the standard approach is to use DP-SGD (Abadi et al., 2016) to inject noise into the gradients computed at every iteration of SGD training, and use composition theorems to bound the total privacy leakage across iterations.

**Differentially Private Prediction** Private prediction differs from private training in that the notion of approximate indistinguishability applies to a sequence of predictions made by a *private prediction protocol* $\mathcal{P}$, rather than to a privately trained model. Formally, at each time step $t$, an adversary Adv (potentially adaptively) issues a context string $\mathbf{x}_t$, and the private prediction protocol $\mathcal{P}$ responds by generating a next token $y_t \in \Sigma$. We let $\mathcal{P} \leftrightharpoons_T$ Adv denote the sequence of query-response pairs between P and Adv up until time $T$: $\mathcal{P} \leftrightharpoons_T \text{Adv} = \{\mathbf{x}_t, y_t\}_{t=1}^T$. For a query sequence of length $T$, approximate indistinguishability requires that for adjacent datasets $\mathcal{D}, \mathcal{D}'$:

$$D \left( \mathcal{P}(\mathcal{D}) \leftrightharpoons_T \text{Adv} \, || \, \mathcal{P}(\mathcal{D}') \leftrightharpoons_T \text{Adv} \right) \leq \epsilon, \tag{1}$$

for some divergence $D$ and $\epsilon > 0$. We summarize the above discussion in the following Rényi-DP variant of the definition for private prediction by Dwork & Feldman (2018).

**Definition 2.3** (User-level RDP for private prediction). Let $\alpha > 1$, $\epsilon > 0$, and $T \in \mathbb{Z}_+$. A prediction protocol $\mathcal{P}$ is $(\alpha, \epsilon, T)$-RDP if for any adversary Adv and any $\mathcal{D}$ and $\mathcal{D}'$ that differ in only one user's data $\mathcal{D}_i$, we have that Equation 1 holds.

It is well-known that differentially private models can be used for private prediction via the *post-processing theorem* (Mironov, 2017): If $h \leftarrow \mathcal{M}(\mathcal{D})$ is a model obtained from an $(\alpha, \epsilon)$-RDP training mechanism $\mathcal{M}$, then $\mathcal{M}'(\mathbf{x}; \mathcal{D}) = h(\mathbf{x})$ is an $(\alpha, \epsilon, \infty)$-RDP private prediction mechanism for any sequence of queries (regardless of length). However, DP-SGD (Abadi et al., 2016)—the primary

mechanism for training private neural networks—makes an implicit assumption that the adversary also observes additional information that is not accessible if $h$ is used as a prediction API, and in practice, it causes the accounting mechanism in DP-SGD to vastly overestimate the privacy leakage parameter $\epsilon$ (Nasr et al., 2021). One alternative is the general-purpose *subsample-and-aggregate* mechanism, which adds noise to an ensemble's output in order to privatize it. This results in a trade-off between the information leakage, $\epsilon$, and the number of queries that can be answered, $T$ (van der Maaten & Hannun, 2020). For smaller $T$, the mechanism needs less noise to achieve a particular $\epsilon$. Conceptually, this is a step in the right direction, but the added noise greatly reduces utility and is superfluous if we can leverage pre-trained public LMs to privatize the predictive distribution.

### 2.3 OPERATIONAL PRIVACY FOR PRIVATE PREDICTION

To remedy the problems in user-level differentially private training and prediction, we propose a different notion of privacy that is sufficient for preventing text extraction attacks, but admits more specialized privacy mechanisms with tighter privacy accounting.

Let $\mathbb{P}(\mathcal{D})$ denote the power set of $\mathcal{D}$. A *partition* $\Pi \in \mathbb{P}(\mathcal{D})$ of $\mathcal{D}$ is a collection of sets $\pi$ that satisfies $\bigcup_{\pi \in \Pi} \pi = \mathcal{D}$ and that satisfies $\pi \cap \pi' = \emptyset$ for distinct $\pi, \pi' \in \Pi$. We refer to the elements of $\Pi$ as *parts*. For some fixed ordering, we let $\Pi_i$ denote the $i$-th part. As a minor abuse of notation, we let $\mathcal{D} \setminus \pi$ denote the usual element-wise subtraction and write $\Pi \setminus \pi$ instead of $\Pi \setminus \{\pi\}$ for brevity. Recall the notion of the private prediction protocol $\mathcal{P}$. We augment the protocol $\mathcal{P}$ with the capability to terminate the query-response sequence at any time. With a slight abuse of notation, we denote by $T(\mathcal{P})$ the sequence length produced by $\mathcal{P}$. We define *Rényi operational privacy* (ROP) as follows.

**Definition 2.4** (Rényi operational privacy for private prediction)**.** Let $\mathcal{D}$ be a dataset of user-level corpora and let $\Pi$ be a partition so that each user-level corpus $\mathcal{D}_i$ is contained in some part $\Pi_j \in \Pi$. For $\alpha > 1$ and $\epsilon > 0$, a prediction protocol $\mathcal{P}$ is $(\alpha, \epsilon)$-ROP for partition $\Pi$ of dataset $\mathcal{D}$ if for any part $\Pi_i \in \Pi$ and adversary Adv, we have:

$$D_\alpha^{\text{sym}} \left( \mathcal{P}(\Pi) \underset{T(\mathcal{P})}{\leftrightharpoons} \text{Adv} \, || \, \mathcal{P}(\Pi \setminus \Pi_i) \underset{T(\mathcal{P})}{\leftrightharpoons} \text{Adv} \right) \le \epsilon.$$

Where $D_\alpha^{\text{sym}}(P||Q) = \max\{D_\alpha(P||Q), D_\alpha(Q||P)\}$. ROP differs from user-level RDP in Definition 2.3 in two aspects:

1. We substitute user-level adjacency with partition-level adjacency. By definition, the partition is constructed so that any user's data belongs to a single part. Readers familiar with *group privacy* (Dwork et al., 2014) may recognize partition-level adjacency as a formal relaxation of the group-level adjacency used in Rényi group differential privacy. Partition-level RDP is neither strictly weaker nor stronger than user-level RDP, and, at a cost, conversion is possible (Appendix C).
2. We allow *variable-length* query-response sequences by enabling the private-prediction protocol $\mathcal{P}$ to terminate[3] at will. ROP accounts for the privacy leakage in the responses made throughout the prediction protocol's operation lifetime, which is why we refer to it as *operational*. By allowing for a variable-length sequence, we provide the mechanism with additional flexibility without increasing susceptibility to text extraction. Non-sensitive queries often can be answered without much privacy leakage, whereas sensitive queries may quickly exhaust the privacy budget, causing the protocol to terminate early. The protocol's decision to terminate at time $T(\mathcal{P})$ may leak some information about how sensitive the queried contexts are. However, this leakage is relatively insignificant and can be upper bounded (allowing us to convert variable-length $\epsilon$ into fixed-length; see Appendix C).

## 3 SUBMIX

We introduce SUBMIX, a private next-token prediction protocol that satisfies the operational privacy definition introduced above ( Figure 1). SUBMIX follows the design of the subsample-and-aggregate mechanism by first forming a random partition $\Pi$ of the training dataset, with each user's data belonging to a single random part $\Pi_i$. For each part $\Pi_i$, the protocol further splits $\Pi_i$ into two *subparts* $\pi_i$ and $\pi_i'$ by randomly assigning users in $\Pi_i$ to the two halves.

---

[3]After termination, the mechanism can technically continue to issue responses, but only in a way that is entirely independent of the private corpus, *e.g.*, by using a public pre-trained LM.

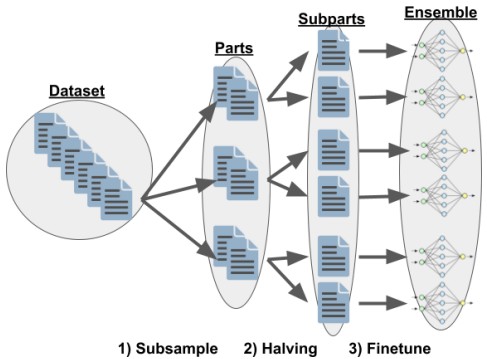

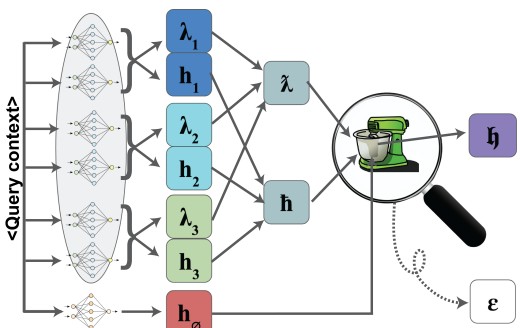

(a) **Training.** The corpus is a dataset comprised of private user text. Each document represents all of the text corresponding to a particular user. At training time, SUBMIX learns an ensemble by: (1) subsampling the dataset into non-overlapping parts, (2) halving each part into two subparts, and (3) fine-tuning the LM on each subpart using $\mathcal{L}$.

(b) **Prediction.** The bottom-most network in the figure represents the pre-trained public model; the other networks form the ensemble of model pairs obtained after SUBMIX training. SUBMIX prediction produces a mixing weight for each model. These weights are aggregated and used to mix the ensemble predictions with the predictions of the public model.

Figure 1: Overview of SUBMIX's training protocol (**left**) and prediction protocol (**right**).

**Fine-tuning a Pre-trained Model** Language models are often first trained on vast internet crawls to develop a general understanding of human language, and then fine-tuned on a more domain-specific dataset for the target task (Dai & Le, 2015; Howard & Ruder, 2018). We treat language model training as a black-box operation, and denote the

---

**Algorithm 1** SUBMIX Training.

**Inputs:** User-level private corpus $\mathcal{D}$, LM fine-tuning routine $\mathcal{L}$
**Outputs:** Fine-tuned LMs $h_{\pi_i}, h_{\pi'_i}$ for $i = 1, \ldots, k$
**Hyperparameters:** # of parts $k$

1: $\Pi \leftarrow$ Random $k$-fold partition of $\mathcal{D}$ with $|\Pi_i| = |\mathcal{D}|/k$
2: **for** $i \in \{1, ..., k\}$ **do**
3: $\quad (\pi_i, \pi'_i) \leftarrow$ Randomly split part $\Pi_i$ into two subparts.
4: $\quad h_{\pi_i} \leftarrow \mathcal{L}(\pi_i), h_{\pi'_i} \leftarrow \mathcal{L}(\pi'_i)$
5: **end for**

---

training routine $\mathcal{L}(\cdot)$ as a function that returns a model $h_{\mathcal{D}} = \mathcal{L}(\mathcal{D})$. We assume access to a LM pre-trained on public data, and use routine $\mathcal{L}$ to fine-tune the LM on the private user-level corpora. Specifically, Algorithm 1 fine-tunes a public pre-trained LM on each subpart $\pi_i$ and $\pi'_i$ to produce LMs $h_{\pi_i} = \mathcal{L}(\pi_i)$ and $h_{\pi'_i} = \mathcal{L}(\pi'_i)$. By convention, fine-tuning on the empty set returns the public pre-trained model: $h_{\emptyset} = \mathcal{L}(\emptyset)$.

**Next-token Distribution** Given a query context $\mathbf{x}_t$, each part $\Pi_i$ is responsible for producing a next-token probability mass function (pmf) by combining $h_{\pi_i}(\mathbf{x}_t)$ and $h_{\pi'_i}(\mathbf{x}_t)$ into $\bar{h}_i(\mathbf{x}_t) = (h_{\pi_i}(\mathbf{x}_t) + h_{\pi'_i}(\mathbf{x}_t))/2$. SUBMIX mixes this pmf with the public pre-trained model $h_{\emptyset}$ to add noise to the prediction that hides private information. It does so by computing:

$$h_i(\mathbf{x}_t) = \lambda^* \bar{h}_i(\mathbf{x}_t) + (1 - \lambda^*) h_{\emptyset}(\mathbf{x}_t),$$

for a suitable choice of the mixing weight $\lambda^*$. A value of $\lambda^* = 0$ means the fine-tuned LMs $h_{\pi_i}$ and $h_{\pi'_i}$ are not used (no privacy loss), and $\lambda^* = 1$ means no noise was added (no utility loss). We select $\lambda^*$ based on how much information about the part $\Pi_i$ is contained in the pmfs $h_{\pi_i}(\mathbf{x}_t)$ and $h_{\pi'_i}(\mathbf{x}_t)$.

Intuitively, since both $\pi_i$ and $\pi'_i$ are random samples from the same data distribution, if the models $h_{\pi_i}$ and $h_{\pi'_i}$ did not memorize the query context $\mathbf{x}_t$ then $h_{\pi_i}(\mathbf{x}_t)$ and $h_{\pi'_i}(\mathbf{x}_t)$ will be similar. Hence, the selected value of $\lambda^*$ should be close to 1. If either $h_{\pi_i}$ or $h_{\pi'_i}$ memorized the context $\mathbf{x}_t$, then $h_{\pi_i}(\mathbf{x}_t)$ and $h_{\pi'_i}(\mathbf{x}_t)$ are dissimilar as $\pi_i$ and $\pi'_i$ have no users in common. This suggests that mixing with the pre-trained LM $h_{\emptyset}$ is necessary for hiding the sensitive information in $\Pi_i$, so $\lambda^*$ should be close to 0. SUBMIX balances between these two extremes by computing a separate $\lambda_i$ for each part $\Pi_i$. Specifically, it sets a target privacy leakage $\beta > 0$ and optimizes:

$$\lambda_i \leftarrow \max_{\lambda \in [0,1]} \{\lambda : \mathsf{D}_i(\mathbf{x}_t, \lambda) \leq \beta\}, \tag{2}$$

where $\mathsf{D}_i(\mathbf{x}_t, \lambda) = D_\alpha \left( \lambda h_{\pi_i}(\mathbf{x}_t) + (1 - \lambda) h_{\emptyset}(\mathbf{x}_t) \,\|\, \lambda h_{\pi'_i}(\mathbf{x}_t) + (1 - \lambda) h_{\emptyset}(\mathbf{x}_t) \right)$. The final value of $\lambda^*$ is obtained by averaging the $\lambda_i$ values for $i = 1, \ldots, k$, where $k$ is the number of parts.

---

**Algorithm 2** SUBMIX Prediction.

---

**Inputs:** Fine-tuned LMs $h_{\pi_i}, h_{\pi_i'}$ for $i = 1, \ldots, k$, privacy parameters $\epsilon$, time step $t$, query context $\mathbf{x}_t \in \Sigma^*$
**Outputs:** Next token response $y_t \in \Sigma$
**Hyperparameters:** Rényi divergence order $\alpha$, target leakage $\beta$

1: **if** $t = 1$ **then**
2:     $\varepsilon_i \leftarrow \epsilon$ for $i = 1, \ldots, k$
3: **else if** STOP has been issued **then**
4:       **return** $y_t \sim h_\emptyset(\mathbf{x}_t)$
5: **end if**
6: **for** $i = 1, \ldots, k$ **do**
7:    $\bar{h}_i(\mathbf{x}_t) \leftarrow \frac{1}{2}(h_{\pi_i}(\mathbf{x}_t) + h_{\pi_i'}(\mathbf{x}_t))$
8:      Compute $\lambda_i$ using Equation 2.
9: **end for**
10: $\lambda^* \leftarrow \frac{1}{k} \sum_{i=1}^k \lambda_i$
11: $\bar{h}(\mathbf{x}_t) \leftarrow \frac{1}{k} \sum_{i=1}^k \bar{h}_i(\mathbf{x}_t)$
12: $h(\mathbf{x}_t) \leftarrow \lambda^* \bar{h}(\mathbf{x}_t) + (1 - \lambda^*) h_\emptyset(\mathbf{x}_t)$
13: **for** $i = 1, \ldots, k$ **do**

14:     $\lambda^*_{-i} \leftarrow \frac{1}{k-1} \sum_{j \neq i} \lambda_j$
15:     $\bar{h}_{-i} \leftarrow \frac{1}{k-1} \sum_{j \neq i} \bar{h}_j(\mathbf{x}_t)$
16:     $\mathfrak{h} \leftarrow \lambda^* \bar{h}(\mathbf{x}_t) + (1 - \lambda^*) h_\emptyset(\mathbf{x}_t)$
17:     $\mathfrak{h}' \leftarrow \lambda^*_{-i} \bar{h}_{-i}(\mathbf{x}_t) + (1 - \lambda^*_{-i}) h_\emptyset(\mathbf{x}_t)$
18:     $\varepsilon_i \leftarrow \varepsilon_i - \max\{D_\alpha(\mathfrak{h}||\mathfrak{h}'), D_\alpha(\mathfrak{h}'||\mathfrak{h})\}$
19: **end for**
20: **if** $\forall i: \varepsilon_i > 0$ **then**
21:     $y_t \sim h(\mathbf{x}_t)$
22: **else**
23:     Issue STOP signal.
24:     $y_t \sim h_\emptyset(\mathbf{x}_t)$
25: **end if**
26: **return** $y_t$

---

**Prediction and Privacy Accounting**    Given the next-token pmfs $h_i(\mathbf{x}_t)$ for $i = 1, \ldots, k$, SUBMIX computes the ensemble pmf, $h(\mathbf{x}_t) = 1/k \sum_{i=1}^k h_i(\mathbf{x}_t)$, and samples from it to obtain a next-token prediction. Our mechanism for selecting the mixing weight $\lambda^*$ can be shown to limit the privacy loss of a sample from $h(\mathbf{x}_t)$ under the operational privacy notion: Since each $\lambda_i$ is determined entirely by the part $\Pi_i$, the next-token pmf after removal of $\Pi_i$ can be derived in closed form. This allows us to compute the Rényi divergence in $h(\mathbf{x}_t)$ for adjacent datasets $\Pi \setminus \Pi_i$. We present the SUBMIX prediction protocol in Algorithm 2, and give its formal privacy analysis in the following proposition.

**Proposition 3.1.** SUBMIX *is an* $(\alpha, \epsilon)$-*ROP prediction mechanism.*

*Proof.* We use the adaptive sequential composition theorem for RDP (Mironov, 2017, Prop. 1). By construction, Algorithm 2 ensures that $\max_i \sum_t D_\alpha (y_t \sim \mathsf{P}(\Pi) \,||\, y_t \sim \mathsf{P}(\Pi \setminus \Pi_i)) \leq \epsilon$ for privacy budget $\epsilon$ and divergence order $\alpha$. Even if the query contexts $\{\mathbf{x}_t\}$ are chosen adaptively by the adversary, Proposition 1 in Mironov (2017) allows us to sum the Rényi divergences from each release:

$$\max_i D_\alpha \left( \mathsf{P}(\Pi) \leftrightharpoons \mathsf{Adv} \,||\, \mathsf{P}(\Pi \setminus \Pi_i) \leftrightharpoons \mathsf{Adv} \right) \leq \max_i \sum_{t=1}^{T(\mathcal{P})} D_\alpha \left( y_t \sim \mathsf{P}(\Pi) \,||\, y_t \sim \mathsf{P}(\Pi \setminus \Pi_i) \right)$$

$$\leq \max_i \sum_t \varepsilon_i(t) - \varepsilon_i(t-1) \leq \max_i \epsilon = \epsilon.$$

To conclude the analysis, note that after SUBMIX issues the STOP signal, any subsequent queries are answered by $h_\emptyset$. Since $h_\emptyset$ is not a function of $\Pi$, this does not leak any additional information. $\square$

## 4   EXPERIMENTS

**Datasets**    We evaluate SUBMIX by fine-tuning the pre-trained GPT-2 model from HuggingFace (Wolf et al., 2019) on two "private" datasets: (1) `Wikitext-103` (Merity et al., 2016), a collection of 103 million tokens scraped from Wikipedia; and (2) `BigPatent-G` (Sharma et al., 2019), a collection of over 200,000 patents. We split the `wikitext-103` corpus into blocks of length 512 tokens and define each block as a (synthetic) user $\mathcal{D}_i$. We split `BigPatent-G` by patent and define each user to be a single patent. This setup mimics settings in which users have distinct data distributions within the text corpus.

**Fine-Tuning & Evaluation**    We use standard hyperparameters for fine-tuning; see Appendix B for details. To assess the quality of LM predictions, we measure predictive perplexity:

$$\mathbf{PP}_h = \mathbb{E}_{\mathbf{x} = x_1 \cdots x_L \sim \mathcal{D}_{\mathrm{heldout}}} \left[ \exp \left( -\frac{1}{L} \sum_{i=1}^L \log h(x_i | x_1 \cdots x_{i-1}) \right) \right],$$

where $\mathcal{D}_{\text{heldout}}$ is the private held-out set and $h(\cdot|x_1 \cdots x_{i-1})$ denotes the pmf for the next token given context $x_1 \cdots x_{i-1}$. In practice, we truncate the context window to a fixed maximum length $L = 512$. Since each held-out sample consists of a block of $L = 512$ tokens, computing $\mathbf{PP}_h$ for a single sample requires $L$ queries in total. Hence, the total number of queries $B$ is a multiple of 512.

Following Geumlek et al. (2017), we report privacy loss in terms of $\alpha$-Rényi divergence. Note that we can convert from $(\alpha, \epsilon)$-RDP to $(\epsilon', \delta)$-DP via $\epsilon' = \epsilon + \frac{\log(1/\delta)}{\alpha-1}$ (Mironov, 2017). In the paper, we measure $\epsilon$ using $\alpha = 2$ Rényi divergence; we present results for other values of $\alpha$ in Appendix A.

## 4.1 Privacy-Utility Trade-off

**Baseline Comparisons** We first compare SUBMIX to three privacy-preserving mechanisms as baselines: (1) DP-SGD (Abadi et al., 2016) for private training; and (2) subsample-and-aggregate (S&A, Dwork et al. (2014)) and (3) GN-Max (Papernot et al., 2018) for private prediction. See Appendix B for details on adapting these mechanisms for private next-token prediction. Figure 2 shows the predictive perplexity of the private mechanisms on the Wikitext-103 dataset for ROP/RDP[4] parameters $\alpha = 2$ and $\epsilon = 2$. The pre-trained GPT-2 model has a perplexity of 37.5 on Wikitext-103 and is trivially private on that corpus. Fine-tuning

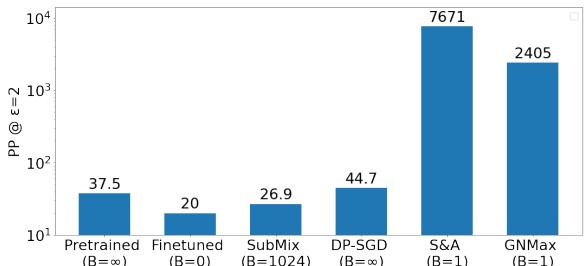

Figure 2: Perplexity for $\epsilon = 2$, $\alpha = 2$ of four privacy-preserving mechanisms (SUBMIX, DP-SGD, S&A, and GN-Max) using GPT-2 on Wikitext-103 with varying query budgets $B$. Non-privately fine-tuning achieves a perplexity of 20.0 and the pre-trained public model achieves a perplexity of 37.5 . Lower perplexity is better.

the LM non-privately achieves a perplexity of 20.0. SUBMIX achieves a perplexity of 26.9 at $B = 1,024$ queries, which is substantially below the perplexity of the pre-trained LM. By contrast, the other mechanisms do not improve over the pre-trained baseline, even for a *single* query ($B = 1$). subsection A.1 presents more detailed results on the privacy-utility trade-off of the baseline methods: all of them require extremely large $\epsilon$ to improve over the pre-trained baseline, with DP-SGD outperforming the private prediction baselines (S&A and GNMax).

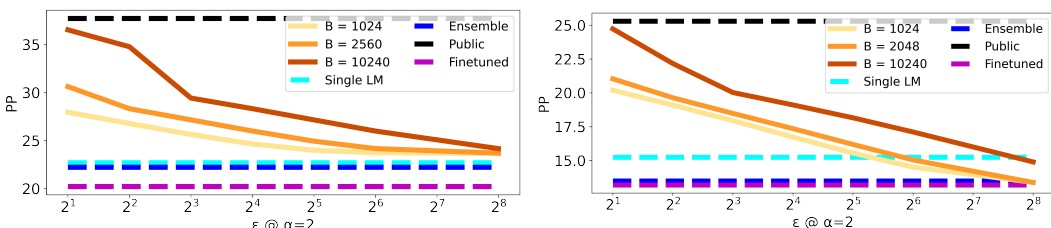

Figure 3: Perplexity of SUBMIX ($k = 8$) on Wikitext-103 (**left**) and BigPatent-G (**right**) as a function of ROP privacy loss $\epsilon$ for three query budget values $B$. The perplexity of the pre-trained model and (non-private) single model, ensemble, and fully fine-tuned models are shown for reference.

**Varying the Privacy Loss** In Figure 3, we show the trade-off between perplexity and ROP privacy loss, $\epsilon$, of SUBMIX at $\alpha = 2$. On both Wikitext-103 (left plot) and BigPatent-G (right plot), SUBMIX substantially improves over the pre-trained GPT-2 model, even when the query budget is increased to $B = 10,240$ queries. As expected, SUBMIX matches the perplexity of non-private LM at higher values of $\epsilon$. Interestingly, SUBMIX's perplexity is even lower than that of a single non-privately fine-tuned LM at high $\epsilon$. We surmise this is due to the performance gap between a single fine-tuned LM and an LM ensemble. The effect is less pronounced on Wikitext-103 because that corpus was split into users by block, as a result of which many LMs contain text blocks from the same Wikipedia article. This reduces the positive effects of ensembling on predictive perplexity.

---

[4]Privacy loss is measured under RDP for DP-SGD, S&A, and GNMax; and under ROP for SUBMIX.

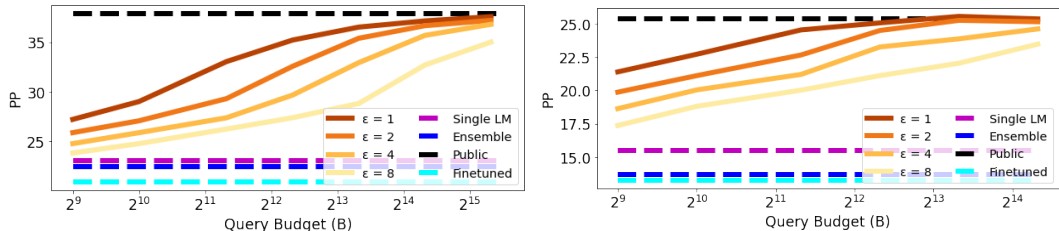

Figure 4: Perplexity of SUBMIX ($k = 8$) on `Wikitext-103` (**left**) and `BigPatent-G` (**right**) as a function of query budget $B$ for different ROP privacy losses $\epsilon$. The perplexity of the pre-trained model and (non-private) single model, ensemble, and fully fine-tuned models are shown for reference.

**Varying the Query Budget** Figure 4 shows the trade-off between perplexity and the number of queries $B$. The results in the figure were obtained by tuning the target leakage to obtain the desired budget. As expected, answering more queries using SUBMIX increases the average perplexity for each next-token query at a given $\epsilon$. However, SUBMIX attains a surprisingly low perplexity for a moderate number of queries (*e.g.*, $B = 2^{10}$) at all $\epsilon$ values on both `Wikitext-103` and `BigPatent-G`.

**Varying the Number of Parts** A key hyperparameter of interest in SUBMIX is the size of the partition $\Pi$. Intuitively, a smaller number of partitions, allows each part to train a better quality LM at the cost of a larger Rényi divergence when a part is removed. Figure 5 shows the trade-off between perplexity and ROP privacy loss, $\epsilon$, for varying partition sizes, $k$. We observe the key trend that as $\epsilon$ decreases, perplexity increases more rapidly for smaller $k$ because the privacy budget is exhausted more

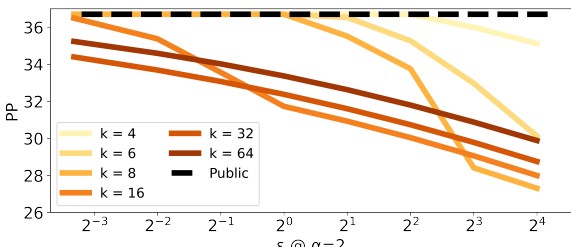

Figure 5: Perplexity of SUBMIX on `Wikitext-103` as a function of ROP privacy loss $\epsilon$ for different partition sizes $k$.

quickly when each part has a greater relative contribution to the ensemble. The optimal value for $k$ is generally around 16 for all $\epsilon$ values of interest, although this may depend on the data distribution and design choices such as model architecture and training hyperparameters.

## 4.2 TEXT EXTRACTION ATTACKS

To empirically validate that SUBMIX prevents text extraction attacks, we perform a random code text-extraction experiment in the style of Ramaswamy et al. (2020); Shi et al. (2021). We randomly generate $m$ codes with each code being an $\ell$-digit number (for example, representing a user's age, ZIP-code, phone number, SSN, *etc.*). The fine-tuning dataset is constructed so that each user's text is single sentence: $\mathcal{D}_i =$ "My number is: <random $\ell$-digit number here>". We then fine-tune on this dataset and make private predictions using SUBMIX for the query

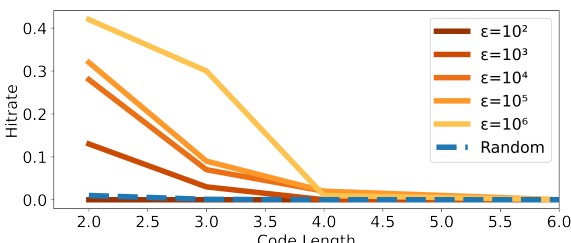

Figure 6: Hit rate of text extraction attacks on SUBMIX for varying lengths of code. The # of parts is $k = 3$ and the # of codes generated is $g = 100$. The non-privately fine-tuned LM has a hit rate of $\geq 0.9$ for all lengths.

context "My number is:" to test whether SUBMIX prevents text extraction (and if so, at what $\epsilon$). As a baseline, we fine-tune GPT-2 on this dataset for 1000 iterations. This results in the LM memorizing all $\ell$ codes, achieving a perplexity of less than 0.5 and $\geq 90\%$ recall when prompted with context. We apply SUBMIX with $k = {}^m/_2$ parts so that each model in the ensemble strongly memorizes one number, achieving near $100\%$ recall when prompted with the context.

For the text extract attack, the figure-of-merit is the *hit rate* of the $g$ generations, defined as the number of generated codes that exactly match a secret code divided by $g$. Figure 6 shows the hit rate of the text extraction attack. For all code lengths $\ell = 2, \ldots, 5$, SUBMIX succeeds in preventing the attack at $\epsilon = 10^2$. For longer code lengths, even higher values of $\epsilon$ suffice for preventing this random-sampling text extraction. Intuitively, extraction becomes more difficult as the code lengths increase. This experiment shows that the mechanism for limiting the release of sensitive information via solving Equation 2 is effective, and the privacy accounting in SUBMIX meaningfully measures the amount of privacy loss. We also ran these attacks for varying values of $K$ and $m$, and made the same qualitative observations. In addition, we performed experiments in which we varied $g \in \{10, 100, 1000\}$. We found that this does not affect SUBMIX's hit rate.

## 5 DISCUSSION & RELATED WORK

**Related Work** McMahan et al. (2017) was the first to study the training of differentially private language models by using DP-SGD to train a small recurrent neural network. However, the resulting LMs have far fewer parameters than modern transformers and attain much higher perplexities. More recently, Shi et al. (2021) explored an alternative approach called *selective differential privacy*, where the privacy guarantee only applies to blocks of text in the training corpus that are deemed sensitive, *e.g.*, addresses and phone numbers. Unfortunately, this approach is difficult to scale to large unstructured text corpora because it requires annotating all text in the corpus with a privacy sensitivity level.

SUBMIX has conceptual similarities to PATE (Papernot et al., 2016; 2018) for private semi-supervised learning. Both SUBMIX and PATE make predictions using an ensemble of models trained (or fine-tuned, in the case of SUBMIX) on private data, and employ a data-dependent and query-dependent privacy accounting mechanism at prediction time. The central idea in both methods is that privacy loss is small when models trained on different parts of the data agree on a prediction. However, PATE is more natural in discriminative or classification tasks because it return a distribution's noisy argmax. On the other hand, SUBMIX is more natural in generative tasks because it returns a sample from the distribution.

SUBMIX is also related to prior work on private posterior sampling (Geumlek et al., 2017; Dimitrakakis et al., 2017), where the randomness in the privacy mechanism comes from releasing a sample from a distribution defined by the private data. In particular, SUBMIX uses a privacy accounting methodology based on Rényi divergences similar to that of Geumlek et al. (2017).

Mireshghallah et al. (2021) propose adding a privacy regularizer to language model training to reduce its memorization of sensitive text. They empirically showed that the regularizer reduces the effectiveness of text extraction attacks, but it does not satisfy any formal privacy guarantee such as DP. Other related works exploring privacy in natural language processing include (Gopi et al., 2020; Lyu et al., 2020; Xu et al., 2020; Li et al., 2018; Kim et al., 2021).

**Limitations & Future Directions** One limitation of the SUBMIX protocol as presented here is that it only supports decoding from the ensemble of LMs by sampling directly from the predicted pmf. This type of sampling is known to produce unnatural and incoherent text (Kulikov et al., 2018; Holtzman et al., 2019). Better decoding methods such as top-k sampling (Fan et al., 2018) and nucleus sampling (Holtzman et al., 2019) exist, but they require modifications to the protocol that may cause additional privacy leakage. However, we note that a close alternative to top-k and nucleus sampling is *temperature decoding* (Holtzman et al., 2019), which scales the predicted pmf by a temperature term to decrease its entropy. SUBMIX readily supports this decoding method by applying temperature scaling as a post-processing step. In future work, we aim to extend out work by designing protocols that can support different types of decoding strategies as well.

Another limitation of SUBMIX is that the use of an ensemble substantially increases the computational and storage requirements. Our experiments suggests that an overhead factor of $8$ is needed to attain a non-vacuous trade-off between privacy and utility. One potential solution to reduce the computational requirements of SUBMIX may be to fine-tune only the top few transformer layers closest to the prediction head. This would allow the evaluation of the bottom transformer layers to be shared between models in the ensemble, thereby reducing both computation and storage requirements. We leave the exploration of such efficiency improvements for future work.

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
