# OpenReview forum: "SubMix: Practical Private Prediction for Large-scale Language Models"
_ICLR.cc/2022/Conference — ICLR 2022 Submitted_

### Official Review · Reviewer_kTn8 · 2021-10-26

**Correctness:** 4
**Technical Novelty And Significance:** 3
**Empirical Novelty And Significance:** 2
**Recommendation:** 8
**Confidence:** 3

**Details Of Ethics Concerns:**

No obvious ethics concerns

**Main Review:**

Pros:
1. Submix does not need to modify training algorithm and existing large-scale language models are available.
2. Submix utilize probabilistic nature of next-token sampling to protect privacy.
3. Submix does not need to add noise for privacy of model prediction.

Cons:
1. Fail to show runtime comparison of proposed approach and existing work (e.g., DP-SGD). Except effectiveness analysis, efficiency analysis may convince reader more since less efficient approach may weaken its application in the real world. Authors can provide a chart with respect to runtime vs. epsilon.

Question:
1. Since this work uses non-private language models for fine-tuning, is it under risk of member inference attack?
2. About user-level corpus, since young people sometimes use similar internet words, phrases and sentences on their social media posts, D_i and D_j sometimes may not be disjoint set. How should Submix deal with this scenario?

**Summary Of The Paper:**

Language models keep sentence-level information of training samples in detail. This feature jeopardizes data privacy by data-extraction attacks, which can retrieve full sentences of training samples by directly querying models. Current solution to this issue is based on differential privacy, such as DP-SGD. However, existing approach suffers from information leakage due to full access to model’s parameter/gradients and only can be applied to small-scale RNNs, which also cannot balance privacy-utility trade-off. In this work, the proposed approach called SUBMIX focuses on private prediction for answering next-token queries with regular language models fine-tuned on the private corpus. Submix as an ensemble based on models by private corpus and public pre-trained models, and only mix them if they disagree. Privacy leakage is measured by Renyi divergence, and operational privacy is guaranteed.

**Summary Of The Review:**

Sentence-level privacy in the language model is a very challenge problem since LMs applied to many tasks in NLP. To this end, Submix provides a scalable and flexible approach to address the issue by focussing on next-token prediction, which is popular for many downstream NLP tasks. Hence, I recommend to accept this paper.

---

> ### Author Response · Authors · 2021-11-14
> **Response from Authors**
>
> Thank you for your time and feedback. We address your concerns below:
>
> *“Fail to show runtime comparison of proposed approach…”:* Regarding computational efficiency, since our method can be highly parallelized for prediction by distributing the parts to different GPUs, in practice it does not carry any significant overhead in wall-clock time.  For fine-tuning, the ensemble training does not incur any compute overhead since each of the 8 models only trains on 1/8th of the data. We will add a discussion section to the draft about this.
>
> *“is it under risk of member inference attack?”:* As we state in the introduction, our method is not designed to offer any privacy protection with respect to the public data used in pre-training. However, for the private corpus used for fine-tuning, prior work (e.g. Yeom et al., 2018) showed that membership inference can be limited by differentially private mechanisms, including SubMix.
>
> *“About user-level corpus, since young people sometimes use similar internet words..”:* See our general response about correlation attacks and memorization.  Our method is aimed at preventing extraction of k-eidetically memorized secrets for small k, hence it remains effective even if users share common tokens.

---

### Official Review · Reviewer_aiFL · 2021-10-30

**Correctness:** 2
**Technical Novelty And Significance:** 2
**Empirical Novelty And Significance:** 2
**Recommendation:** 3
**Confidence:** 4

**Main Review:**

I applaud the authors for focusing on the problem of privacy in language modeling, however, I have some major concerns about the assumptions made and the actual applicability of the proposed method, that I will list below:

1. Faulty assumptions in the experiments: The paper has defined their setup to be "finetuning" a pretrained language model. In finetuning tasks in NLP, it is often the case that training data is scarce. However, here for the experiments, Wikipedia is used as "private" dataset, which is far from a realistic scenario, especially when there are many realistic available datasets that NLP/privacy researchers use, such as Enron email dataset,  sentiment 140, Shakespeare, etc (I have listed multiple related missing privacy in NLP reference papers below, there are more appropriate datasets there as well). I strongly believe that if a small dataset is used, the utility of the proposed method will also suffer gravely, given how it partitions the data and leaves few training samples for each large model. I think an experiment on a small dataset could clear the outcome of this issue.

2. Assumption on the correlation between users: I highly disagree with the assumption that if all the LM partitions agree on the next word for a given sequence, then there is no leakage. There is the possibility that users are correlated, like in an email dataset in a company, and a large group of them might all be talking about a secret, like a secret serial number. If different users talk about this, there is a chance that each partition ends up with at least one user that has that serial number in their data. Therefore, they might all predict it, and agree on it, and this could leak that string. Please clarify on how such a scenario can be prevented in your setup.

3. Huge computation/storage costs: Although the limitations section of the paper mentions this and proposes using a smaller model as a solution, I highly disagree. The proposed method's functionality depends on prior pretraining and then finetuning. This is not the usage mode for small LSTM models. So the proposed solution isn't really feasible.

4. The entire NLP-related body of work on this problem is missing. This is not  just a few missing references. I have provided a list below:


1. Gopi S, Gulhane P, Kulkarni J, Shen JH, Shokouhi M, Yekhanin S. Differentially private set union. InInternational Conference on Machine Learning 2020 Nov 21 (pp. 3627-3636). PMLR.

2. Mireshghallah F, Inan HA, Hasegawa M, Rühle V, Berg-Kirkpatrick T, Sim R. Privacy Regularization: Joint Privacy-Utility Optimization in Language Models.

3. Lyu L, He X, Li Y. Differentially Private Representation for NLP: Formal Guarantee and An Empirical Study on Privacy and Fairness. InProceedings of the 2020 Conference on Empirical Methods in Natural Language Processing: Findings 2020 Nov (pp. 2355-2365).

4. Xu Z, Aggarwal A, Feyisetan O, Teissier N. A Differentially Private Text Perturbation Method Using Regularized Mahalanobis Metric. InProceedings of the Second Workshop on Privacy in NLP 2020 Nov (pp. 7-17).

5. Li Y, Baldwin T, Cohn T. Towards Robust and Privacy-preserving Text Representations. InProceedings of the 56th Annual Meeting of the Association for Computational Linguistics (Volume 2: Short Papers) 2018 Jul (pp. 25-30).

6. Kim K, Gopi S, Kulkarni J, Yekhanin S. Differentially Private n-gram Extraction. arXiv preprint arXiv:2108.02831. 2021 Aug 5.



**Summary Of The Paper:**

The paper attempts at introducing a privacy-preserving method for predicting next words, for large language models. The problem setup for the paper is when pretrained large language models are available, and they are to be fine-tuned on private data for the task of next word prediction. If DP-SGD is used here, due to the large number of parameters in these models, the authors claim that the noise that is added would be too much and utility (i.e. perplexity) would suffer gravely. Therefore, they introduce an ensemble-based approach: partitioning the data into disjoint subsets, such that each user's data appears in only one partition. Then, finetuning a separate model on each partition. For decoding (i.e. inference, guessing the next word), if all the partition-LMs agree, then they assume that there is no leakage and release it. If there is disagreement, the output of these models is mixed, along with the output of a publicly pretrained LLM, so as to bound the private leakage (kind of similar to PATE, but for language modeling). Then the paper proceeds to attempt at empirically testing this method on two datasets.

**Summary Of The Review:**

To summarize the points from above, I believe the main shortcomings of this paper are:

1. unrealistic assumptions in the experiments and about the problem setup in general (relating to the size of data and correlation between users).
2. Huge computation and storage costs which are not explicitly studied.
3. Not providing any comparisons with prior existing work in NLP.

If the authors show that this method is effective in a setup where the user data is scarce, address my concern on corrolations, and provide comparisons (qualitatively) with prior work I am willing to update my score.

---

> ### Author Response · Authors · 2021-11-14
> **Response from Authors**
>
> Thank you for your time and feedback. We address your concerns below:
>
> *“Faulty assumptions in the experiments…”:* We respectfully disagree on the usage of small datasets. Differential privacy relies on the ability to “blend in the crowd”, which inherently requires large-scale datasets. This is a very reasonable requirement for modern applications since there is often a vast availability of unstructured private data that can be used to finetune a language model. For example, when using private emails and messages to train an autocompletion model, there could easily be millions of users each with a significant amount of text. In our experiments, both WikiText-103 and BigPatent contain roughly 200K users (synthetic users in the case of WikiText), which is a moderate number.
>
> *“Assumption on the correlation between users…”:* Please see our general comments and updates concerning eidetic memorization and correlation attacks.
>
> *“Huge computation/storage costs:”* We would like to clarify regarding the resource usage of SubMix. We used an ensemble size of 8 outside of our ensemble size ablation (Fig 5), which implies an 8X overhead in storage cost and computation at prediction time. However, this can be easily addressed by parallelizing the models across GPUs in a node at prediction time, hence wall-clock times remain virtually the same. Furthermore, we anticipate that this overhead factor remains constant for larger language models, hence it is manageable with system optimizations. For fine-tuning, the ensemble training does not incur any compute overhead since each of the 8 models only trains on 1/8th of the data.
>
> *“The entire NLP-related body of work on this problem is missing….”:* Thank you for pointing out this related work. We added a discussion that contains these references in Section 5 of our revised draft. Although these papers are relevant, only Mireshghallah et al is tackling a comparable problem, but their notion of privacy is empirical and hence does not admit a fair comparison to our work.

---

> > ### Comment · Reviewer_aiFL · 2021-11-22
> > **Author Response**
> >
> > Thank you for the response. Your response answers my last two concerns, thank you. However, the following two concerns remain:
> >
> > “Faulty assumptions in the experiments…”: The fact that differential privacy relies on a lot of data does not mean it is there for all cases, especially for fine-tuning a large language model (and not training it). Fine-tuning is the act of adapting a pre-trained model to a new domain, given a small data. If fine-tuning data size was actually wikipedia size, then it wouldn't really be fine-tuning, you could just train on that from scratch. Fine-tuning data domains could be Avocado email dataset, or MNLI/QNLI.
> >
> > “Assumption on the correlation between users…”: I agree that this problem exists in DP too, but that doesn't make it go away here. Just because something is a problem with DP does not make it ok. Also, this is not technically a 'problem' with DP, since DP is to be applied to cases where there are no correlations, and again, this does not seem to be the case here.

---

> > > ### Author Response · Authors · 2021-11-23
> > > **Response to Reviewer aiFL from Authors**
> > >
> > > Thank you for taking the time to read and consider our responses. We’d like to respond again below.
> > >
> > > “Faulty assumptions in the experiments…”: We would like to emphasize that the dataset GPT2 was pre-trained on is an 40GB internet corpus (https://cdn.openai.com/better-language-models/language_models_are_unsupervised_multitask_learners.pdf), while Wikitext-103 is only about 500MB, and so GPT2’s pre-training data is still ~80X larger. As described in https://arxiv.org/abs/2108.07258, these large pre-trained models can serve as a “foundation” to build different applications and are not limited to fine-tuning on small datasets. Furthermore, the point of the pre-trained model is to give SubMix a data-independent reference language model that has good “general” language model capabilities. As we explained in our paper, this is important because mixing the prediction with a general-purpose LM results in a far better perplexity than adding random noise to the prediction. The “fine-tuning” aspect of the SubMix protocol is not relevant for this, and, if it performed better, the LM training could be initialized from scratch instead of from the GPT2 pretrained weights. From making predictions, SubMix could still use the pretrained GPT2 in this case as well.
> > >
> > > “Assumption on the correlation between users…”: As we mentioned in our general response, DP is sufficient to protect against correlation and eidetic memorization among a small group of users. As you’ve pointed out, the focus of DP-style guarantees is to prevent such scenarios since the extraction of secrets pertaining to a small subset of users can compromise personal privacy, which is very harmful to participants in the dataset. As we’ve referenced in the paper, there is a prior line of research focusing on DP for language models, so we feel it has been established as a worthwhile privacy notion for LMs. We agree with you that when there are confidential messages between users within a large organization (i.e. corporate or government secrets), the possibility of a large-scale secret is real. One SubMix-compatible solution is to combine all of the training data from any single such “secret-possessing” organization into one “combo” user (i.e. combine all emails within “@bureau.gov” or “@corp.com” into one user). In this case, we are sure that all of that data ends up in the same partition, and thus the partitions cannot  “agree” on the prediction of these secrets. In most cases, these “secret-possessing” organizations will be using end-to-end encryption for all of their important communications, and encrypted text is useless for training an LM anyways.
> > >
> > > We will be certain to add both of these explanations to our manuscript in order to clarify these issues. We hope you find that our clarifications alleviate your concerns!

---

### Official Review · Reviewer_zjmQ · 2021-11-03

**Correctness:** 1
**Technical Novelty And Significance:** 1
**Empirical Novelty And Significance:** 1
**Recommendation:** 3
**Confidence:** 5

**Main Review:**

First, the reviewer does not understand why the proposed approach defends against the text extraction attacks without releasing the trained models? The prediction setting provided in the proposed approach may be suitable for ML as a service (MLaaS). However, why next token is an MLaaS application and in which real-world scenarios? Also, the connection between user level-DP with text extraction attacks is weak. Why is user level-DP needed to defend against text extraction attacks?

Second, the threat model is missing. The reviewer is not sure how the adversaries will choose the query $x_t$ to extract sensitive text data. It is unclear from the writing.

Third, the user level-DP provided in the proposed approach by partitioning users’ text data into different parts and sub-parts is incorrect. For the fine-tuning models to work well, data in a single part must be sufficiently large, requiring a certain number of users included in the part. As a result, the neighboring part definition (excluding one part) does not provide a precise user level-DP. We can only have user level-DP if a user is a part. This scenario will not work well for fine-tuning large NLP models, such as GPT-2, given a limited amount of text data of a user.

Also, the assumption that if two models return similar results when they do not memorize the context of a query $x_t$ is very vague. It is unclear to me when this could happen and how we can quantify the similarity here. No evidence was provided. The same issue given the two models return dissimilar results when they memorize the query $x_t$ given that they do not share common users.

Fifth, experimental results are unconvincing. DP-SGD does not provide either user level-DP or solutions to prevent text data extraction attacks. Why is DP-SGD being considered a baseline here? At least, the user level-DP from McMahan et al. (2017) and its variations (Kairouz et al., 2019; Ramaswamy et al., 2020) should be considered as appropriate baselines. Improving the perplexity in the fine-tuning does not indicate that the model is secure against data extraction attacks.

Regarding the attack evaluation, why randomly generating $m$ codes with each code being an $l$-digit number (for example, representing a user’s age, ZIP-code, phone number, SSN, etc.) can be a representative measure for information leakage under data extraction attacks? Sensitive information can be extremely broader than these numbers, such as name entity, correlation among them, etc. What would be a good way to evaluate the model under the attacks and under which threat models?

**Summary Of The Paper:**

This paper presents a new approach to simultaneously preserve user level-DP and prevent text extraction attacks in prediction. Experimental results were conducted on Wikitext-103 and BigPatent-G datasets showing some promising results. However, there are several critical concerns detailed in the main review regarding the application contexts, theory analysis, and experimental settings.

**Summary Of The Review:**

Overall, this is an important research direction. The proposed approach is interesting. However, there is room for improvement.

---

> ### Author Response · Authors · 2021-11-14
> **Response from Authors**
>
> Thank you for your time and feedback. We address your concerns below:
>
> *“The prediction setting provided in the proposed approach may be suitable for ML as a service (MLaaS)...”:* While private prediction certainly applies to MLaaS, it is applicable in numerous centralized settings as well. Consider an autocorrect or phrase completion system for an email client. A model pretrained on internet data could benefit from fine-tuning on user data, although maintaining the privacy of a user’s emails is obviously a concern. In this setting, the model can stay on the server while the autocompleted texts are forwarded to the client. The requirement that the model stays server-side may appear as a drawback compared to private training, but in fact, large transformer models are unsuitable to be deployed on client devices due to limitation of computing resources.
>
> *“Why is user level-DP needed to defend against text extraction attacks?”:* Thank you for pointing this out. Please refer to our general response for a formal treatment of this issue.
>
> *“Second, the threat model is missing…”:* The threat model is the same as that in Dwork & Feldman. We’ve clarified this more explicitly in Section 2 in the revision. Sorry for the confusion.
>
> *“Third, the user level-DP provided…”:* We apologize for the confusion surrounding the partition-level DP and thank you for pointing this out. We hope our updated manuscript can address these concerns (see general response).
>
> *“Also, the assumption that if two models return similar results when they do not memorize …”:* We believe there is a misunderstanding regarding this statement. To clarify, this is a heuristic that we used as a guiding principle in the design of SubMix, while its formal privacy guarantee is given using rigorous Renyi DP accounting. Loosely speaking, if the next-token pmfs between two models are extremely similar, then an adversary cannot make reliable inferences about any text that’s unique to the training data of either model. Such a heuristic is a recent but well-established privacy concept (for example, see PATE) that can be shown formally to amplify privacy. In our case, similarity is quantified by the Renyi divergence between the next-token pmfs, as we explain in Section 3 and in our code sketch. Our method precisely keeps track of this Renyi divergence to control the total information leakage and is not just a heuristic.
>
> *“Why is DP-SGD being considered a baseline here?...”* We would like to clarify our definition of users in the experiments. For the comparison on WikiText-103 in Figure 2, we simulated users by splitting up the entire corpus into blocks of 512 tokens, with each block representing a user (this is mentioned in Section 4). Each block is treated as a single sample during training, so user-level privacy is equivalent to sample-level privacy. Note that this definition of users is as favorable as possible to DP-SGD since it maximizes the number of users. We will add this clarification to the revision.
>
> *“Regarding the attack evaluation, why randomly generating…”:*  We agree that text extraction attacks are broader than our experiment, which is why we opted for the rigorous privacy notion of DP and supplemented our privacy analysis and evaluation with the random code extraction experiment as a sanity check. It is also worth noting that our random code extraction experiment followed the setup in prior works such as Ramaswamy et al. (2020) and Shi et al. (2021).

---

### Official Review · Reviewer_6t5J · 2021-11-04

**Correctness:** 3
**Technical Novelty And Significance:** 3
**Empirical Novelty And Significance:** 2
**Recommendation:** 6
**Confidence:** 2

**Main Review:**

The paper is well written and experiments show that the method works in practice. The visualizations are also great. The paper studies a difficult problem as utility of next word prediction suffers a lot when trained with differential privacy.

Experimental section does not provide details on whether other methods with DP were trained using an already pre-trained models and not only using private corpora. This needs to be made clear that the improvement in the utility is not only from the user of a pre-trained model (for example, why would one even use DP-SGD and not the pretrained model?)

The comparison with PATE seems a bit brief as the methods are quite alike albeit PATE was used for classification tasks. It would be interesting to see how SUBMIX performs on classification tasks.

Please explain while splitting into two parts is needed after the data is already split in k.

Minor: Fig 1 b calls the second model differently than in Alg 1 and 2 (h and lambda)

**Summary Of The Paper:**

The paper proposes a method for answering next-token queries in a privacy-preserving manner. As opposed to training models with DP guarantees, it adds privacy-preserving guarantees during the prediction time. The approach works by getting a public model and then fine-tuning it on disjoint parts of the private dataset. This way it obtains, several models. During prediction, it queries each model and combines the outputs by proportionally adjusting the contribution from each model. The experiments how that utility is better than that pf DP training and also protects against extraction attacks of sentences.

**Summary Of The Review:**

The paper presents an interesting idea for privacy-preserving prediction for fine-tuned models. However, as written now it is not clear if the benefit in experiments is due to fine-tuning or due to the new method of training/prediction proposed in the paper.

---

> ### Author Response · Authors · 2021-11-14
> **Response from Authors**
>
> Thank you for your time and feedback. We address your concerns below:
>
> *“Experimental section does not provide details on whether other methods with DP…”:*  We can confirm that DP-SGD was used to fine-tune the same pretrained model used in SubMix.
>
> *“The comparison with PATE…”:* With regards to SubMix and PATE, we agree there are some important conceptual similarities between the methods. However, PATE is designed to output the noisy argmax of a distribution, whereas SubMix is designed to sample from the distribution. Therefore, PATE is a more natural fit for a classification task compared to a generative task. Nevertheless, we agree that exploring SubMix (and variants of SubMix) on different tasks and applications is an interesting direction for future work.
>
> *“Please explain while splitting into two…”:* Regarding splitting into two subparts: In SubMix, each part is responsible for outputting a mixing parameter lambda_i that is adaptively tuned to how sensitive the part’s next-token prediction is to its training data. Conceptually, if the part’s prediction is the same on any random draw of training samples then its sensitivity is low. However, since we do not have access to additional training samples, we simulate it in a way similar to bootstrapping by subdividing the part, treating each subpart as the random draw of training samples to the other subpart. We will clarify this in the revised draft.
>
> *“Minor: Fig 1 b calls…”:* Thank you for pointing out the typographical error. We will address this.

---

### Author Response · Authors · 2021-11-14
**General Response to Reviewers from Authors**

We thank the reviewers for their valuable time and feedback. We have updated our draft in response to their suggestions. In the following summary, we discuss the main changes and how they relate to the reviewer feedback.

*Connection between DP and text extraction.* Several reviewers questioned why differential privacy prevents text extraction. In response, we added a formal treatment in appendix D to explicitly state this connection. We showed that the DP privacy parameter epsilon upper bounds the success rate of extracting eidetically memorized text, where eidetic memorization is defined in Carlini et al. (2019). Although this connection has been implicitly assumed in prior work, to the best of our knowledge it has never been formally proven, hence it constitutes as an additional technical contribution of our paper.

*Correlation attacks.* Reviewers aiFL and kTn8 mentioned a type of correlation attack, where a similar piece of text is shared across multiple users, hence prevention of 1-eidetic memorization is not sufficient. Using group DP, any DP guarantee still provides some privacy for texts shared among a small number of users, but the privacy level decreases as the number of users in the group grows. Thus, DP can still prevent k-eidetic memorization for small k but not large k. We added a discussion of this issue in appendix C in the revision. However, we emphasize that vulnerability to correlation attack is a well-known weakness of DP; see https://arxiv.org/abs/1603.01508. While we believe that it is important to devise new privacy definitions to address this concern, such a consideration is out of scope for our paper.

*Partition-level vs. user-level privacy.* We opted for the notion of partition-level privacy because it is a natural alternative to user-level privacy that greatly simplifies data-dependent privacy accounting. Since partitioning is done at the user level and not at the sample level, exclusion of a part is equivalent to exclusion of every user in that part, and should be no weaker than user-level privacy. In appendix C of the revised draft, we give conversions between partition-level and user-level privacy with some additional accounting overhead. This result formally validates our claim that partition-level privacy is neither stronger nor weaker than user-level privacy, and hence it is reasonable to choose either one as the formal privacy notion.

---

### Decision · Program_Chairs · 2022-01-20

**Decision:**

Reject

**Comment:**

This paper develops a new mechanism SubMix that provides next-token prediction under a variation of the differential privacy constraint. There is disagreement among the reviewers when assessing the quality of this work. Even though the study of private predictions in large language models is new, the reviewers raised several issues in the proposed approach. First, the formulation of partition-level DP created confusion about the privacy guarantees provided by the mechanism. Given the similarity to PATE, it might be useful to articulate if there is any difference between the privacy guarantee in this paper and the one of PATE. Second, the authors might want to further clarify the reason for having two sub-parts, which has also created some confusion. Even after reading the author's response and the updated revision, the AC still could not understand the relevant privacy argument. In summary, the paper may require further clarification and revision before it is ready for publication.